# The Emergence and Impact of Ethylene Scavengers Techniques in Delaying the Ripening of Fruits and Vegetables

**DOI:** 10.3390/membranes12020117

**Published:** 2022-01-20

**Authors:** Mohd Affandy Aqilah Mariah, Joseph Merillyn Vonnie, Kana Husna Erna, Nasir Md Nur’Aqilah, Nurul Huda, Roswanira Abdul Wahab, Kobun Rovina

**Affiliations:** 1Faculty of Food Science and Nutrition, Universiti Malaysia Sabah, Jalan UMS, Kota Kinabalu 88400, Sabah, Malaysia; bn17110034@student.ums.edu.my (M.A.A.M.); vonnie.merillyn@gmail.com (J.M.V.); mn1911017t@student.ums.edu.my (K.H.E.); aqilah98nash@gmail.com (N.M.N.’A.); drnurulhuda@ums.edu.my (N.H.); 2Department of Chemistry, Faculty of Science, Universiti Teknologi Malaysia, Johor Bahru 81310, Johor, Malaysia; roswanira@utm.my

**Keywords:** food packaging, fresh produce, shelf life, packaging applications

## Abstract

As the top grocery list priorities, the primary challenge when purchasing fruits and vegetables from supermarkets is obtaining fresh, minimally processed perishable goods. This source of diet is critical for obtaining vitamins, minerals, antioxidants, and fibres. However, the short shelf life caused by moisture content in rapid deterioration and decay caused by microbial growth, results in unappealing appearances. Fruits and vegetables undergo ripening and eventually the ageing process, in which the tissues of the plants degrade. Even after harvesting, numerous biological processes occur, generating a significant variation of ethylene production along with respiration rates between fruits and vegetables. Thus, the utilization of ethylene scavengers in food packaging or films has been revealed to be beneficial. The synergistic effects of these biomaterials have been demonstrated to reduce microorganisms and prolong the shelf life of greens due to antimicrobial activity, oxygen scavenging capacity, enzyme immobilization, texture enhancers, and nutraceuticals. The current review fills this void by discussing the most recent advances in research on ethylene scavengers and removal mechanisms of ethylene, including oxidation in fruit and vegetable packaging. The application and advantages of ethylene scavengers in packaging are then discussed with the addition of how the efficiency related to ethylene scavengers can be increased through atmospheric packaging tools. In this context, the article discusses characteristics, types of applications, and efficacy of ethylene control strategies for perishable commodities with the inclusion of future implications.

## 1. Introduction

Consumers in developed countries are keen to learn about the manufacturing processes and ingredients used to make the food they consume. They are becoming more conscious of food quality, particularly in respect to purchasing farm-fresh products to maintain wellness. Additionally, buyers make purchasing decisions about food based on its physical characteristics, as fresh produce is exceptionally susceptible to quality and physical appearance issues during storage [1]. After harvest, most fruits and vegetables release ethylene, one of the simplest phytohormones. Ethylene initiates ripening, causes chlorophyll softening and degradation, with the ultimate result of the deterioration process of fresh commodities. This phytohormone accelerates the ripening and senescence of fresh produce, impairing the quality of its products, essential minerals, and commercial viability. The gas may initiate ripening by degrading chlorophyll, resulting in deterioration [2]. This is due to ethylene gas being ubiquitous in the air and contributes to post-harvest food loss that must be minimized during transportation, stockpiling, and management. [3,4].

Environmental factors consist of mostly the temperature, humidity, and packaging atmosphere which have an effect on the horticultural crops, post-harvest physiology, and chemical composition during storage. There are numerous methods for reducing ethylene production or inhibiting its action for preserving the quality and shelf life of produce. This is through active and intelligent food packaging innovation, whereby the food industries have developed an interest in using fewer ingredients due to customer encouragement [5]. Advancement on intelligent systems continuously monitor food storage conditions, expiration dates, quality, safety, microbial growth, and fresh food. Utilizing active packaging with ethylene scavenging attributes can mitigate ethylene’s unpropitious consequences and remove it from the atmosphere [6]. In this case, the active packaging system modifies the environment, in which packaged food is stored, during the preservation period prior to maintaining food safety and preserving quality [7].

Alternative materials for ethylene elimination are now commercially available, including silica gel, activated alumina metal oxides, layer silicates, clays, zeolite, and activated carbon, packaged in sachets and inserted into packaging or incorporated together with the packaging material [2,6,8,9,10]. The impregnation inside the packaging matrix with ethylene scavenging agents rather than adding them directly to the package in sachet form has caused a reduction in the amount of ethylene scavenging agents required while still meeting customer demands for safe and healthy food. The demand for environmentally friendly packaging materials has also accelerated due to concern for global climate change and by that, recent studies on incorporating this scavenging agent into packaging made from waste materials such as pine needles has been carried out and possessed remarkable ethylene scavenging properties [10]. Moreover, packaging that incorporates ethylene removal systems has been shown to significantly delay the ethylene’s deterioration effects during storage [11,12]. As a result, multiple efforts have been made to slow the rate at which fresh produce degrades due to ethylene, including the use of ethylene absorbers, ethylene oxidizing agents, ethylene inhibitors based on chemicals, nanomaterials, and immobilized methods [13,14].

This review illustrates the different ethylene scavengers/inhibitors and the evolving techniques for removing ethylene that have been developed in recent years through a particular emphasis on ethylene scavengers/inhibitors that contain catalysts to enhance in situ oxidation. We discuss emerging technologies that are capable of reducing gas permeability and ethylene absorption. Despite the widespread use of ethylene scavengers/inhibitors containing catalysts, most notably metal oxide catalysts, they have not been frequently reported in packaging films, and the purpose of this review was to fill that void. The practice of encapsulating ethylene scavengers, including zeolite, titanium dioxide, and transition metals in a small sachet has been widely reported. However, incorporating ethylene scavengers into food packaging substances, as well as in situ ethylene oxidation, has received relatively little attention.

## 2. Active and Intelligent Packaging Systems

As the global population grows, the requirements for adhering to food safety guidance are constantly updated, along with stricter food safety regulations, numerous food products are introduced to the market, and consumers must continuously monitor their quality until consumption. The desire to keep food as contaminant-free as possible during preservation prompted the improvement of several advanced packaging systems, including intelligent and active packaging systems. Food must be packaged to preserve and maintain its quality prior to export, storage, or consumption. Customer preferences for fresh, organic ingredients and ready-to-eat produce with a long shelf life have increased, causing the development of sophisticated packaging technology.

With regard to innovative packaging technologies, such intelligent and active packaging are becoming more prevalent in the food industry, while many are still being developed and have not yet reached commercialization. Intelligent systems improve safety and alert users to potential hazards in the food packaging atmosphere [15]. Intelligent packaging benefits include fault detection, quality control, and monitoring packaged food products, from manufacturing to consumption, by utilizing indicators and a range of sensors, such as time, temperature and gas indicators, and humidity with an implementation of biosensors. On the other hand, active packaging prolongs the shelf life of goods by absorption and diffusion of different gases, including carbon dioxide, oxygen, and ethanol. There are several critical concerns about these emerging technologies, which include their expenditure, marketability, consumer acceptance, food safety, organoleptic properties, and environmental protection. As a result of these issues, future research is highly recommended to confront them and advance the aspect of intelligent and active packaging in the food industry [16].

Intelligent packaging is defined as the capability of a food packaging method to monitor and communicate changes from intrinsic and extrinsic environmental stipulations. The packaged food interacts to keep supply chain stakeholders such as producers, retailers, and consumers informed of the system’s status. Intelligent packaging is a relative innovation in food packaging because of the significant potential to improve food security [17]. Intelligent packaging elements may improve reliability and contribute to performance enhancement by providing information related to potential pitfalls. Intelligent packaging necessitates are the sensors and recognition systems that allow the estimation and communication of a food product to users [18]. Active packaging is designated as the packaging with subsidiary components that are purposefully incorporated into or on the packaging material or headspace of the package in order to boost the effectiveness of the package policy [5]. Several active packaging systems are available nowadays, including gas scavengers, flavor absorber/emitter systems, antioxidants, antimicrobials, and humidity controllers. The moisture aspect affects the texture, taste, presentation, and microbial interaction of food materials. Scavengers and diffusion practices contain chemicals, flavors, antioxidants, and antimicrobials, whereby these are among the most widely used active packaging technologies for the purpose on lengthening the shelf life of products [19,20,21,22]. The mechanism of intelligent and active packaging for the food industry are summarized in Figure 1.

## 3. Biosynthesis Cycle and Implication of Ethylene on Fruit and Vegetable Ripening

Ethylene is a phytohormone that involves the regulation of numerous physiological functions in plants. A high ethylene concentration is critical for regulating the harmful effects of climacteric fruits and vegetables, including crumbling, overmature fruit, expedited quality deterioration, augmented susceptibility to fruit pathogens, and physiological disorders, which occur primarily during post-harvest and storage [23,24]. There are two types of ethylene that affect ripening, depending on their sources, endogenous ethylene, which is produced biologically by the plant and external ethylene, which is generated by adjacent crops, automotive exhaust, polymers, and tobacco [25]. Additionally, ethylene can induce the expression of genes associated with ripening via the signal transduction pathway [26].

Endogenous ethylene is attributed directly to food with regard to security, flash floods, physiological injuring, foodborne pathogen disease, chemical causative agents, alongside specific stages of growth and development, including germinating seeds and ripening, and flower senescence together with defoliation [27]. As illustrated in Figure 2, endogenous ethylene biosynthesis processes convert methionine to S-adenosylmethionine, which responds with pyridoxal phosphate to form a Schiff base. After removing the hydrogen and substituent from the Schiff base, it transforms into a cyclopropane ring and 5′-methylthioadenosine. The cyclopropane ring is transformed to 1-aminocyclopropane-1-carboxylate, a critical step in synthesizing ethylene, which is then oxidized by ACC oxidase to yield ethylene. Similarly, a 5-methylthioribose molecule converts 5′-methylthioadenosine to methionine, effectively employing the methionine cycle [28,29,30].

Endogenous ethylene production is more complicated than exogenous ethylene synthesis, which is introduced by endogenous developmental signals such as plant development, maturation, and oxidative stress, which happens to be prompted by various exogenous biotic and abiotic exposures [31]. Currently, ethylene production is typically regulated by preventing ethylene synthesis, absorption, oxidation, and adsorption during the handling and distribution of fresh products [25]. Ethylene is especially harmful to fruit and vegetables, whereas even at 0.1 μL/L concentrations it has a significant effect on plant growth and development [28]. Based on the poll released by Keller et al. [25], the adverse effect of ethylene on fresh goods has been projected to result in notable declines of up to 80%, which highlights the importance of speeding up the process of conducting additional research to mitigate this effect. As a result, ethylene control measures, such as ethylene scavengers are critical for minimizing product loss and preserving food quality during post-harvest ripening mechanisms [5].

### 3.1. Ethylene Scavenging Systems

The use of active packaging with ethylene scavenging characteristics aids in mitigating ethylene’s detrimental effects [6]. Based on minimizing ethylene-induced deterioration, ethylene scavenging methods help to extend the shelf life of fresh foods [32]. In theory, ethylene scavenging systems comprise of a simple sachet containing an appropriate scavenging chemical, or an ethylene scavenger incorporated into the composite packing material. Ethylene gas should freely diffuse through the sachet material [33,34]. Scavengers of ethylene are classified into two types: ethylene absorbers, which physically absorb and retain ethylene molecules in their environment, and ethylene scavengers, which absorb water using a synthetic mixture of two materials [6]. The latter is frequently used in packaging, sachets, and films for fresh produce. Although most commercial ethylene scavengers for high-ethylene-activity fresh produce are based on potassium permanganate, the fresh produce industry is increasingly involved on implementing innovative ethylene scavenger components in plastic film-type packaging. The range of materials suitable for fresh products packaging is constrained by cost, safety, and ethylene scavenging capabilities [8].

Numerous alternative ethylene scavenger materials are available nowadays, including activated alumina metal oxides and carbon, zeolite, silica gel, layer silicates, and clays packaged in sachets and placed in packaging or incorporated into the packaging material [2,6,9]. The consequences of reducing the amount of ethylene gas surrounding fresh produce are delaying fruit maturation, and minimizing deterioration, whereby the shelf life of the fresh produce was substantially increased. Producers and commercial enterprises employ equipment to capture the ethylene gas that fresh produce exhales as it reaches maturity while being transported for logistic purposes by truck, plane, or train [35]. These devices effectively inhibit the ripening process, allowing vegetables to be delivered to the market looking fresh and unwilted. For more than two decades, producers and the food industry have safely used these ethylene gas accumulating or neutralizing devices in conjunction with a volcanic ash carrier medium [5].

Recently, Joung et al. [36] employed halloysite nanotubes (HNTs) as an encapsulating source to develop a potassium permanganate-based ethylene scavenger. The researchers discovered that cherry tomatoes wrapped in 1% P-HNTs/LDPE nanocomposite film (P-HLNF) and deposited at 20 °C for 21 days had a low-level rate of ethylene production and respiration, as well as a delay in firmness loss and color change. Besides, Duque et al. [37] ran an experiment to observe the eugenol that affected main quality characteristics of Raf tomato fruits. The preliminary testing revealed that eugenol aids in the preservation of fruit quality after harvest. The fruits were wrapped in a sachet containing a 10:1:1 mixture of clinoptilolite clay, powdered clove buds, and activated charcoal while tested against a commercial ethylene scavenger made mostly of KMnO_4_-impregnated sepiolite. Ground clove buds inhibited ethylene production, which was linked to a delayed maturation. According to the findings, they could be a valuable alternative for active fruit packing in horticulture products. As a result, ethylene scavengers in packaging are potentially applied to prevent ethylene production and thereby, prolong the shelf life of fresh produce while maintaining the quality.

### 3.2. Ethylene Scavenging Mechanism

Fresh produce is now available at a higher price due to an increase in consumer awareness in response to its quality, including smell, color, texture, size, and shape. Fresh produce suffers from physical appearance and quality deterioration when stored because it is susceptible to loss over time [1]. As a result of these efforts, advanced methods are applied to various commercial products to ensure quality and freshness [38]. Packaging systems that remove ethylene can significantly reduce ethylene deterioration during storage [11]. Therefore, many techniques are being developed to slow down the rate of ethylene-induced deterioration of fresh produce [14,39]. Fruit and vegetables are attributed to approximately 50% of wasted materials, making ethylene scavenging materials particularly effective at increasing shelf life and decreasing waste [40].

A good perception of the ethylene scavenging mechanisms yields precise erudition in applicability, marketability, and training systems to control the ethylene gas in the packaging headspace. The appearance of materials that can accumulate, absorb, or covalently change ethylene in ethylene-removing packaging mitigates the effect of ethylene on produce [41]. These active ingredients are designed to prevent or minimize the occurrence of ethylene inside the package, which leads to an increase in production (autocatalysis) or direct changes to the product. Ethylene can be produced within the packaging or diffused outside [42]. Meanwhile, since gaseous ethylene contains a double bond, there is the potential chance that the highly reactive chemical can be altered or destroyed in multiple ways [43]. This creates a plethora of business opportunities for the removal of gaseous ethylene. Technical details such as scavenging capability and rate, mechanisms, derivatives, potential interactions with different active packaging systems and fresh products are considered when developing promising technologies, regulation, and advertising concerns such as cost and food security, together with customer perceptions. Scavenging ethylene can be accomplished chemically or physically [33]. These methods use certain materials or the ability of different treatments to oxidize, degrade, or absorb ethylene gas via different techniques. The most frequently used chemical oxidizer of ethylene is potassium permanganate (KMnO_4_), which is widely embedded in various materials as porous sachets to enhance scavenging capability. The oxidizing reaction of the KMnO_4_ is able to change ethylene gas to acetate and ethanol, without affecting the fresh produce. The mechanism by which KMnO_4_ oxidizes ethylene is based on the alkene oxidation theory. As a result, the ethylene molecule’s carbon–carbon double bond is attacked by KMnO_4_ via electron-rich oxygen termini, causing it to oxidize in the presence of water [32,42]. KMnO_4_ oxidizes ethylene to form acetaldehyde, converted to acetic acid, carbon dioxide, and water. The redox reaction changes the color of KMnO_4_ from purple (MnO_4_ ions) to brown (MnO_2_) [42].

Additionally, the ethylene gas can be scavenged by inhibiting the ethylene receptor. The ethylene blocking system relies on volatile compounds capable of obstructing ethylene’s binding sites resulting in the ethylene being unable to bind to its produced receptor. This method is based on the ethylene being inactivated and oxidized in the packing atmosphere [23]. Commercially available inhibitors include aminoethoxy vinyl glycine (AVG), silver thiosulfate, and 1-methyl cyclopropane (1-MCP). The most frequently used commercial volatile ethylene inhibitor is 1-MCP, inhibiting ethylene binding sites in fresh food products [44]. Ethylene adsorption on activated carbon has received considerable attention since it utilizes natural waste raw materials. Activated carbon (AC) can be synthesized from virtually any carbonaceous material. AC has low inorganic matter, simplicity of activation, accessibility, low cost, and low degradation [45]. AC’s performance as an ethylene adsorbent is influenced by its surface chemistry, surface area, pore volume, temperature, and relative humidity [42]. Granular, powdered, and fiber AC are the most constantly used methods of ethylene removal. The particle size of AC affects its ability to eliminate ethylene due to changes in surface area, porosity, and activation effectiveness [46]. The Langmuir isotherm for ethylene adsorption on granular AC indicates that ethylene forms a monolayer on the adsorbent surface [47]. The pore size of AC is critical in the adsorption process, as ethylene requires a pore diameter greater than 3.9 (ethylene’s kinetic diameter) [42]. Additionally, photocatalytic oxidation can remove ethylene from fresh produce when titanium dioxide (TiO_2_) is used as a catalyst. All parameters affecting ethylene elimination via photocatalytic oxidation with TiO_2_ include relative humidity, temperature, crystal size/phase, and ethylene and O_2_ concentrations [48]. TiO_2_ can be modified to enhance its degradability of ethylene. TiO_2_ oxidizes ethylene photo catalytically to produce carbon dioxide and water [49]. This generates reactive oxygen species that are capable of oxidizing ethylene to carbon dioxide and water, whereby TiO_2_ must be activated in the presence of oxygen and water with UV radiation along with a wavelength approaching 380 nm [50]. It is unknown precisely how the process related to TiO_2_ chemically degrades ethylene.

## 4. Classification and Preservation Mechanisms of Ethylene Scavengers

There are four aspects involving the fresh-keeping mechanisms of fruit and vegetables. One of the aspects of fruit and vegetables ripening is inhibited or slowed by certain chemicals. When the amount of ethylene in a solution is more than 0.01 μL/L, this boosts the respiratory intensity of fruit and vegetables, which speeds up the destruction of chlorophyll, resulting in nutritional depletion in parallel with the end of shelf life. The second aspect is inhibiting bacterial growth in fruit and vegetables. Extrinsic factors such as air, high humidity, and high temperature increase the chances of microbial growth and spoilage when storing vegetables and fruit. Thus, it is essential to follow proper food handling procedures to avoid bacterial growth on food. The next aspect is that fresh produce is put into hibernation via modified atmosphere packaging (MAP), which reduces their respiratory intensity. The synergistic effects of low O_2_ and high CO_2_ are utilized to resist the ripening impact of ripening hormones such as ethylene, resulting in lower nutrient consumption and a longer life span for fruit and vegetables. The last aspect that involves the fresh-keeping mechanisms of fruit and vegetables is anti-fog. Since many packaging films are hydrophobic, the evaporation of fruit and vegetables can cause water vapor to condense on the film’s surface, generating water droplets that are not conducive to be seen from outside the film, and allowing germs to reproduce. Thus, anti-fog chemicals must be added in these situations [28]. Ethylene has a regulatory function through modulating biosynthesis and receptors. The biosynthetic route, fruit cell growth, and biofilm permeability are all regulated by ethylene. Liu et al. [51] have studied ethylene’s synthetic routes and ripening mechanisms in plants. The removal of ethylene by inhibitor can be separated into two categories depending on its source such as plant-level activities and environment-level actions. Genetic alterations and chemical treatment mainly control endogenous ethylene inhibition, whereas exogenous ethylene removal is primarily controlled by physical adsorption, oxidation, catalysis, and biofiltration [52].

### 4.1. Methylcyclopropene as an Ethylene Inhibitor

In climacteric fruits and vegetables, the ethylene inhibitor 1-methyl cyclopropane (1-MCP) is frequently used for post-harvest management [53]. A formation of a double bond with the receptor metal, 1-MCP has inhibited the transmission of the ethylene signal and directly prevents ethylene production [54]. 1-MCP, on the other hand, can only suppress ethylene production to a limited extent and cannot wholly eliminate exogenous ethylene. The inner surface of the adsorbent is frequently used to reduce the exogenous ethylene due to numerous closed and interconnected pores [55]. Adsorbents are frequently used with catalysts or oxidants to degrade and remove adsorbed ethylene [56]. This approach can improve the storage and transportation of most fruit and vegetables without compromising their qualitative characteristics [57].

### 4.2. Zeolite as Ethylene Adsorbent

Numerous clays were investigated for their potential as ethylene adsorbents. The effort has been dedicated to using zeolite as an adsorbent in industrial and agricultural applications. The term “zeolite” derives from the Greek words ‘zeo’ (to boil) and ‘litos’ (stone). The current nomenclature and classification of materials derived from zeolite are determined by the International Zeolite Association’s (IZA) Structure Commission, which assigns each material to a three-letter mnemonic code. For example, natural zeolite clinoptilolite is designated as HEU [58]. Zeolite is a hydrous mineral with a rigid anionic framework that can consist of most well networks and porosities. Cavities are composed of exchangeable metal cations, can accommodate neutral guest molecules, and are capable of being removed and replaced [59]. Zeolite’s microporous structure is due to the group of naturally occurring or synthetic crystalline aluminosilicates. This nanoclay is widely used in the purification and filtration processes, whereas it is hugely beneficial for removing heavy metals and radioactive materials [60]. Additionally, active surface materials such as nanoclays (clinoptilolite, pumice, and tobalite) can be used to absorb the phytohormone ethylene physically. These clays could be integrated into packaging applications through extrusion in plastic films [14] or by impregnating paper-based composites [10].

Zeolite-based ethylene scavengers are suitable for commercial applications, particularly in the agro-food industry, due to their nontoxic, abundant, economic, and environmentally friendly properties [61]. Natural zeolite, doped with copper and zinc cations, effectively excludes ethylene and delays the ripening of tomato fruits. However, because natural zeolite promotes tomato ripening, its single use should be reconsidered. Hence, incorporating copper and zinc cations into zeolite support is a novel, emerging post-harvest technology for delaying fruit ripening and potentially increasing market potential for fresh-market tomatoes. The ability of zeolites to exchange divalent transition metal cations for compensating cations enhances their adsorption and catalytic properties [62,63]. When silicon atoms in the zeolite framework are replaced with aluminum atoms, the surface becomes negatively charged. An additional metal cation or hydroxyl proton must be added to balance the charge, resulting in weak Lewis acid or Bronsted solid acid sites, respectively [64].

### 4.3. Ethylene Catalytic Oxidants

Potassium permanganate (KMnO_4_), ozone (O_3_), and titanium dioxide are examples of ethylene catalytic oxidants (TiO_2_). Because of its great oxidizing capacity, purple KMnO_4_ is the most extensively employed external ethylene scavenger, decomposing ethylene via the reactions of 3CH_2_CH_2_ + 2KMnO_4_ + H_2_O → 2MnO_2_ + 3CH_3_CHO + 2KOH, CH_3_CHO + 2KMnO_4_ + H_2_O → 3CH_3_COOH + 2MnO_2_ + 2KOH, and 3CH_3_COOH + 8KMnO_4_ → 6CO_2_ + 8KOH + 2H_2_O [65]. These processes are triggered by moisture from fresh product transpiration and respiration, changing the color of KMnO_4_ from purple to brown, implying the efficiency of KMnO_4_-based C_2_H_4_-scavenging in fruits and vegetables preservation [5]. Taking into account that porous materials have a high specific surface area, integrating KMnO_4_ in them adequately removes ethylene and aids in the removal of undesirable by-products [51]. One of the difficulties with KMnO_4_ is maintaining its effectiveness, which typically reduces due to saturation over time. As a result of the extended storage periods, KMnO_4_-based C_2_H_4_-scavengers must be replenished regularly for high ethylene-producing commodities. Additionally, because of its increased toxicological and natural deep purple color, KMnO_4_ is not frequently integrated into food contact surfaces of packaging because of the possible risk of migration onto fresh foodstuffs [27].

### 4.4. Potassium Permanganate

The oxidation of ethylene to carbon dioxide and water causes exogenous ethylene to be removed. With regard to this, the decrease in MnO_4_ to MnO_2_ revealed that the ethylene scavenger KMnO_4_ is typically used to induce a color shift from purple to brown [66]. As the toxicity of KMnO_4_ shows an inability to incorporate it directly into the product, it is currently applied in sachets to silica gel and alumina materials. While these systems increase the surface area of ethylene contact and ensure its safety, they are space and time intensive. Recent research has concentrated on developing ethylene-scavenging packaging materials based on potassium permanganate as coatings, impregnates, or composites with packaging polymers [67]. According to Khosravi et al. [67], the low crystallinity of the film will result in the direct loading of potassium permanganate. Despite numerous warnings about its potential for human toxicity, KMnO_4_ has been the most predominantly used ingredient to control the levels of ethylene in packaging headspaces due to its cost effectiveness and simplicity [68].

### 4.5. Halloysite Nanotubes

Halloysite nanotubes (HNTs) are aluminosilicates similar in structure to kaolinite that have hollow tubular structures with a diameter less than a micron [69]. The positively charged inner lumen aids in the loading of negatively charged molecules into the HNTs by rejecting the outer surface [70]. HNTs could be rapidly loaded with active factors to create active surfaces and unique geometries. Additionally, previous research has demonstrated that incorporating HNTs into a polymer matrix enhances the polymer matrix’s properties and stability [69,71] and the degree of crystallinity of the film [9].

## 5. Ethylene Scavenger in Film-Based Packaging

Scavengers embedded in film-based packaging are conceivably known to be an excellent solution to packet-related packaging, which can be combined with a solid incorporated into plastic, or used as a surface on the packaging [72]. Nonetheless, research into incorporating an ethylene absorber/scavenger into packing film is limited. According to previous research, a zeolite molecular sieve can be used as an ethylene scavenger that improves the gas permeability of packaging films because of the crystalline porous 3D arrangement [32]. According to Yildirim et al. [32], ethylene scavengers can be incorporated into active packaging by layering, immobilizing, or altering the subtract surfaces, affecting the quality of food products. Additionally, Malshe and Malshe [73] developed coated nano/microparticles in various shapes to meet food demand and other distribution networks while preserving freshness, extending shelf life, and minimizing environmental impact. Numerous products containing ethylene adsorbents or ethylene removal-based preservatives have been marketed. Recently, Freshness Plus Bags are a commercially available silver catalyzed bag that assists in eliminating ethylene gas and decelerating the aging process without compromising nutritional content. Moreover, Lifeline Technologies developed nano silver-based treatments, whereas Debbie Meyer green utilized zeolites to absorb ethylene and had a limited shelf life due to zeolite saturation.

Recently, a TiO_2_-coated polypropylene film was introduced for photocatalytic ethylene oxidation, which significantly extended the shelf life of tomatoes. Packaging requires thin films, which has prompted nanotechnology in film goods [74,75]. Additionally, Li et al. [76] reported combining nanopowders to reduce ethylene in packaged vegetables while they also discovered that incorporating nanoparticles into polyethylene film has significantly reduced fruit softening and browning after day 12. During day 12, an essential parameter of the Chinese jujube browning rate was reduced, indicating that nanopowders composite film is beneficial for fruit quality preservation. Furthermore, Hu et al. [77] studied the impact on active packaging, composed of nanosilver, nano-TiO_2_, and montmorillonite, on the reliability of fully grown kiwi fruit treated with ethylene after 42 days at 4 °C. The findings indicate that kiwi fruits showed weight decline, softening, color change, and a decrease in Brix degrees, although the exact shelf life was not specified.

Boonruang et al. [78] examined the impact of four different types of packaging for the preservation of ‘Nam Dok Mai’ mangoes. These comprised of (i) nonperforated highly gas-permeable film (HNP), (ii) nonperforated ethylene-absorbing highly gas-permeable film (HNPE), (iii) microperforated highly gas-permeable films (HMP), and (iv) standard nonperforated polyethylene film (LNP). The shelf life of mangoes was extended to 40 days when HNPE was used, 35 days when HNP was used, 30 days when HMP was used, and 5 days when LNP was used, compared to 20 days when no packing film was used at 12 °C. The study shows that HNPE films containing zeolites of the MFI type help to delay the ripening of mangoes. Additionally, Esturk et al. [79] investigated the effects of Tazetut (an inorganic product containing 50% different zeolite) on broccoli floret preservation in low-density polyethylene (LDPE) bags. The results revealed that broccoli could be stored at 4 °C for up to 20 days in composite films containing the scavenger, but for only five days in its unpackaged state. After five days of storage, the oxygen concentration in the LDPE bags without an ethylene scavenger was reported below 1%, indicating that they are susceptible to anaerobic fermentation growth. As a result, the development of composite packaging sheets capable of absorbing ethylene while remaining germ-resistant has emerged towards a critical trend. Li et al. [80] successfully developed a packaging film with ethylene scavenging and antimicrobial properties using LDPE as the base material, ZSM-5 zeolite as the ethylene adsorbent, and TiO_2_ as an antimicrobial agent and photocatalyst.

The composite films are opaque and grainy, which present significant disadvantages, including insufficient ethylene capture efficiency due to a lack of active agents that can be incorporated into packaging film without impairing the film’s mechanical properties [5]. Yildirim et al. [32] reviewed the literature and concluded that numerous petroleum-based polymers comprising active chemicals (LDPE, polyvinyl chloride) [9,78,81,82], have also been thoroughly documented for fruit and vegetable preservation. Food packaging companies are looking for natural source-based and biodegradable packaging to reduce their reliance on petroleum-based materials while providing high-quality meals with minimal chemical preservatives. Malhotra et al. [83] discussed the suitability of natural polymers for these applications. Cellophane and paper made from biodegradable lignocellulosic resources are commonly used in food packaging. Besides, chitosan is the second most widely used biopolymer after cellulose that is also largely used for packaging food products due to its antibacterial properties [84]. Polylactic acid is one of the most widely used biopolymers with good biocompatibility including being environmentally friendly in nature [85].

Previously, Peelman et al. [86] applied PLA and a cellulose-based multilayer package to pack cheese, steak, sausage, and pre-fried fries at 4 °C using modified atmospheric packaging (MAP). The strong gas barrier provided by cellulose film, particularly the oxygen barrier, may be the key to extend the shelf life of MAP-packed foods. Paper products are also ideal for food packaging because they are made entirely of cellulose fibers. In comparison to polyolefin-based plastics, paper packaging allows for adequate ventilation, prevents items from drying out, is cost-effective, and has a high degree of mechanical strength. In the presence of ethylene scavengers, antimicrobial agents, or other functional active compounds, paper products can increase their efficacy in preserving fruits. A 4% Bi_2_WO_6_–TiO_2_ (BT) was used to fabricate starch nanocomposite films for ethylene oxidation [3], increasing the rate of ethylene breakdown to 12.4%. Previously, Kaewklin et al. [87] used chitosan and a nanoscale TiO_2_ nanocomposite coating to promote ethylene photodegradation and prolong the ripening process of tomatoes.

Considering the ethylene scavenging impact, it is crucial that the observed film’s antimicrobial properties are set to achieve the best preservation effect. ZnO nanoparticles are often used as one type of antimicrobial agent in chitosan and carboxymethyl cellulose nanocomposite used to preserve soft white cheese [84]. The augmented bacterial barrier provided by ZnO nanoparticles results in an increase of up to 30 days in the shelf life of the cheese at 7 °C. Antimicrobial agents should be effective against a broad spectrum of bacteria, including Gram-positive, Gram-negative, and fungi. Lavoine et al. [88] developed an antimicrobial material for the surface coating of paperboard that was employed to preserve pork liver using chlorhexidine digluconate and micro fibrillated cellulose. In comparison to polyethylene-coated paperboard, the hybrid coated paperboard preserved the freshness of pork liver. Liu et al. [89] coated the paper’s surface with cellulose nano crystals/silver/beeswax composites. Ni et al. [90] coated filter paper with a composite film containing zinc oxide nanoparticles, carboxymethyl cellulose, and guanidine-based starch to create an antibacterial paper that inhibited E. coli well. These findings suggest that this novel method could produce multi-functional paper with enhanced antibacterial and water resistance properties.

The combined effect of ethylene scavenging, and antibacterial enhancement can significantly increase the shelf life of fruit and vegetables, thereby maximizing preservation. Siripatrawan and Kaewklin’s [91] study on TiO_2_/chitosan nanocomposite films demonstrated this. The films revealed tremendous promise as active packaging for post-harvest applications. The composite film CS-TiO_2_-BPPE (black plum peel extract) was also used as an antioxidant, an ethylene scavenger, an antibacterial, and pH-sensitive food packaging material [92]. Da Rocha Neto et al. reported the development of antimicrobial packages containing ß-cyclodextrin (ß-CD) inclusion complexes (ICs) and essential oils of *Cymbopogon martinii* (palmarosa) (ICP) or star anise (ICsa) [93]. Antimicrobial packaging extended the shelf life of apples by 12 days, resulting in a decrease in *Penicillium expansum* growth. Pan et al. [94] recently investigated the use of guanidine-based polymers to create antimicrobial polysaccharide-based materials that can be used to create antimicrobial cellulose- or paper-based goods for fruit and vegetables’ packaging. Guanidine-based polymers outperform standard quaternary ammonium salt or phosphonium-based polymers in terms of minimal inhibitory concentration. Antimicrobial packaging based on guanidine-modified starch is easily expandable [95]. As a result, incorporating an ethylene scavenger into the packaging film along with guanidine polymer is a brilliant idea.

## 6. Forms of Ethylene Scavengers for Packaging Applications

### 6.1. Gelatine-TiO_2_-Coated Expanded Polyethylene (EPE) Foam Nets

Recently, Fonseca et al. [96] investigated the efficacy of gelatin-TiO_2_-coated polyethylene foam nets to delay climacteric fruit ripening. This one-of-a-kind composite design features an ethylene-degrading photocatalytic coating composed of gelatin and titanium dioxide (TiO_2_) immobilized on extended polyethylene (EPE) foam nets. During transportation and storage, inert materials such as EPE foam nets are frequently used to protect the fruit from physical damage while allowing for gas exchange. Under controlled conditions, artificial ethylene photocatalytic breakdown kinetics were investigated using gelatin-TiO_2_ films and gelatin-TiO_2_-coated EPE foam nets. When exposed to UV-A radiation for 12 h, papaya covered in empty foam nets (control research) produced the highest amount of ethylene. However, ethylene generation from papayas wrapped in gelatin-TiO_2_-coated EPE foam nets was lower (4.6 nmol kg^−1^s^−1^) than the control fruit (15.6 nmol kg^−1^s^−1^) during the photocatalysis experiment.

Additionally, papaya climacteric peaks decreased, as evidenced by CO_2_ production and low O_2_ intake rates after 24 h. According to these results, gelatin-TiO_2_-coated EPE foam nets were highly effective at degrading ethylene and preventing its accumulation in the catalyst. Additionally, the papayas reduced the amount of ethylene generated via autocatalysis. Enzymatic conversion occurs in the membrane of plant cells and regulates the production of ethylene [97]. The ethylene concentration within the intracellular environment (endogenous ethylene) increases first, accelerating the fruit’s respiration rate. Exogenous ethylene diffuses from plant tissues to the environment (endogenous ethylene), where it accumulates, stimulating the synthesis of autocatalytic enzymes. Exogenous ethylene particles bind to the structures of protein receptors and coordinate the synthesis of enzymes that catalyze the chemical reactions that occur during fruit ripening.

Under four days of difficult storage conditions, gelatin-TiO_2_ nanocomposite coated polyethylene foam nets efficiently degraded ethylene and delayed papaya ripening. Fruit photocatalyzed with TiO_2_ had decreased ethylene production and respiration rates during the climacteric peak, increased firmness, and better retention of orange pulp, green peel, sweetness along with acidity balance, and no scalds or surface fungi development. The photocatalytic degradation of ethylene inhibited the development of enzymes that catalyze the breakdown of substances such as chlorophyll, pectin, citric acids, and malic acid, which are presented in higher concentrations at the start of papaya stage maturation. Additionally, the foam nets nanocomposite may have acted as a second UV-A light blocker, limiting the amount of UV-A light absorbed by the peel cells.

### 6.2. Chitosan Film

Chemically inert materials, such as titanium dioxide (TiO_2_), are found in several foods, pharmaceuticals, and biological products. When exposed to ultraviolet light, TiO_2_ generates reactive oxygen species (ROS) and hydroxyl radicals on its surface, which react with organic molecules to produce antibacterial and photodegradation of ethylene activities. Antimicrobial activity on the fruit surface are as a result of: contact with the photocatalytic nanocomposite surface; simplification of the ethylene degradation system; lower photocatalyst saturation on the support surface; versatile application; and biopolymers which are a low-cost material derived from renewable resources [48]. Maneerat and Hayata [74] reduced ethylene in packed tomatoes using ethylene photocatalytic oxidation with TiO_2_ coated with polypropylene film. It was demonstrated that tomato shelf life extended when it was introduced with a mesoporous TiO_2_/SiO_2_ nanocomposite, such as ethylene photodegradation [98]. The feasibility of using a chitosan-TiO_2_ nanocomposite film as an ethylene scavenger to prolong the quality and lifespan of tomato fruit was investigated using TiO_2_ (CT) nanocomposite films and chitosan as ethylene scavengers for post-harvest handling.

The photocatalytic activity of the nanocomposite chitosan-TiO_2_ was determined by determining the ethylene content of fruit. Tomatoes packaged in nano-sized titanium dioxide (CT) in chitosan film had significantly lower ethylene concentrations throughout storage (*p* < 0.05) compared to those packaged in chitosan (CS) film and the control, implying that the nanocomposite CT film could photodegrade ethylene and thus delay tomato ripening. Chitosan film degrades ethylene at a much slower rate than CT nanocomposite. Chitosan’s ability to scavenge ethylene is most likely due to the ability of hydroxyl and amino groups in its molecular chain to oxidize. The chitosan-TiO_2_ nanocomposite film outperformed the chitosan film in terms of tensile strength, photodegradation of ethylene, and barrier properties. The nanocomposite CT film that exhibited ethylene photodegradation, retarded ripening, and increased the lifespan of tomato fruit, indicates that ethylene aids in initiating and regulating quality changes in climacteric fruit. The nanocomposite CT film produced can be used to handle fresh food post-harvest.

### 6.3. Sachet

A sachet is a well-known marketable ethylene scavenger. It is based on packaging solutions that scavenge ethylene, whereby this is frequently used to minimize or remove ethylene levels in future product containers. Ethylene scavengers are essentially packaged in a porous sachet and are available in powder, granules, and beads. Limited quantities of steel wire mesh ethylene scavenging sachets are available to supplement the variety of blankets and porous slabs to minimize ethylene in cold storage and transportation containers. Taking account of safety concerns, KMnO_4_ is the most enduring scavenging substance used in sachets. In extreme conditions, moisture should not be directly exposed to powdered ethylene scavengers. Among other factors to have an effect on the size of the sachet required for packing, are the type of fruits and vegetables, the amount of time required to retain the fresh product, the weight of the fresh product, the size of the packaging to be protected, and the onset of gas ethylene sensitivity. The sachets’ primary function is to eliminate ethylene from the distribution supply chain and ensure consistent preservation from the packaging line to the store’s backroom. It is not recommended to use such sachets with high ethylene-producing fresh food for extended storage periods, as the ethylene scavenger may become saturated quickly, necessitating the regular replacement of the sachet [99].

Controlling ethylene and its effects on fruit ripening enables more precise and consistent estimation for pre-climacteric fruit harvest and storage in an ethylene-free environment [100]. Potassium permanganate (KMnO_4_) in sachets or impregnated in plastic containers or chemical filters, oxidizes and absorbs ethylene to form water, manganese dioxide, potassium, and carbon dioxide [101]. On the other hand, it has been demonstrated that eliminating ethylene with potassium permanganate inhibits the ripening of climacteric fruits such as bananas, papayas, and apples. KMnO_4_ sachets impregnated with vermiculite were placed inside each container [102].

### 6.4. Polymeric Films

A variety of polymers and mixtures is sufficient to allow the producer to select the most suitable food packaging solution and configuration for each product. When compared to the transpiration rates of fresh produce, the majority of polymeric films (polyethylene (PE), polypropylene (PP), polyvinyl chloride (PVC), and polyethylene terephthalate (PET)) with a thickness of 10–60 μm are commonly used in fresh produce packaging revealed to have a high water vapor transmission rate (WVTR). Therefore, due to the general lack of familiarity with sachets, ethylene scavenging packaging in polymeric films is gaining traction. Numerous ethylene-scavenging films have been developed using PE and finely dispersed powdered materials such as zeolite, clays, and activated carbon. At the moment, these ethylene scavenging compounds can be incorporated into packaging materials in two ways, which is by melting them into the polymeric matrix or by coating them.

Typically, the incorporation technique was dictated by the type of packaging used to pack fresh fruit and the manufacturer’s ability to incorporate the ethylene scavenger into its products. Previously, Siripatrawan and Kaewklin [91] used a solvent casting technique to fabricate films from chitosan-titanium dioxide nanocomposite. Singh and Giri [103] synthesized KMnO_4_ nanoparticles embedded in SiO_2_ crystals using a low-density polyethylene (LDPE) film, whereas Tas et al. [9] synthesized halloysite nanotubes and polyethylene-based nanocomposites films via an extrusion process. While these functional ethylene scavenger films containing clays physically adsorbed ethylene, their scavenging efficiency was low due to the limited amount of clays that could be incorporated into the polymer matrix without impairing the packaging barrier and mechanical properties [104]. Additionally, as clays are opaque and coarse, they affect the film’s gas barrier, allowing CO_2_ to escape more quickly and O_2_ to enter more quickly than standard plastic films [99].

According to the research made by Tas et al. [9], in comparison to pure polyethylene (PE) films, nanocomposite HNT/polyethene (HNT/PE) films demonstrated an increased ethylene scavenging capacity, and decreased oxygen including water vapor transfer rates. On the strength of their ethylene scavenging properties, nanocomposite films have been found to retard the ripening process of bananas and maintain the firmness of tomatoes. The ability of HNT/PE films containing 5% HNT to scavenge ethylene was demonstrated by the increased shelf life of bananas and tomatoes packaged in these films. Additionally, strawberries and fresh-cut chicken samples wrapped in HNT/PE films containing 1% HNT exhibited superior oxygen and water vapor barrier properties, as demonstrated by the films’ extended shelf life. HNTs are used as natural, nontoxic, and cost-effective nanofillers, which enable the development of potentially multifunctional active food packaging materials that contribute significantly to food safety. Previously, Ahn et al. [105] synthesized and described gallic acid and potassium carbonate as a non-metallic OSS in an LDPE film. On the other hand, aggregation of the OSS resulted in a rough film surface, increased water vapor and oxygen permeability, and decreased the TS. The oxygen scavenging film may scavenge oxygen in the presence of moisture. A 20% concentration of non-metallic OSS was more effective at scavenging oxygen than other samples. The OSS can be used to actively package items with a high moisture content, such as meat and poultry products, fresh-cut fruits, fruit juices, and beer.

## 7. Applications of Ethylene Scavengers in Fresh Produce Packaging

Fresh produce quality has a great deal of influence on the post-harvest process, including physical damage, distribution, and storage management. The latter is crucial with the aim of delivering a good quality product, for the purpose of longer shelf life, particularly for ensuring the quality for buyers. As far as storage management is concerned, ethylene production and accumulation also with temperature and initial gas concentrations of the storage environment must be acknowledged in tandem with maintaining high relative humidity rates for extending the shelf life of greens [104]. In general, the ethylene gas produced by bananas has a significant influence on banana ripening, resulting in green bananas turning yellow and dark brown as they ripen. Therefore, bananas are tough to preserve in good shape and freshness for a long time. There have been numerous studies on this subject over the years, and there is evidence that the uses of ethylene scavengers can improve the packaging of this fruit. Presently, Ezati et al. [106] have prepared carboxymethyl cellulose (CMC)-based functional films using titanium dioxide and copper-loaded titanium dioxide to prevent the browning of bananas. The effect of CMC films on browning inhibition in bananas was investigated by wrapping and storing the bananas for 14 days at 25 °C under visible light. The surface of the control banana and those packed with neat CMC and CMC/TiO_2_ films were dark brown, with a distinctively discolored black and brown tint. On the other hand, the ripening surface color wrapped in CMC/Cu–TiO_2_ film had a better visual appearance. Ebrahimi et al. [107] extended the shelf life of bananas by 15 days by using nanosilica and nanoclay with potassium permanganate. Tzeng et al. [108] extended the shelf life of bananas with a zeolite-based scavenger modified with palladium. Zeolite modified with palladium significantly delayed banana ripening and improved fruit firmness and peel color. Tas et al. [9] designed novel halloysite nanotubes/polyethylene (HNT/PE) nanocomposites as an ethylene scavenging film and neat PE film, which revealed the banana packaged in the NEAT PE film turned yellow, and some brown spots appeared. Bananas wrapped with HNT/PE films, inversely were devoid of brown blotches and preserved their green color.

The presence of ethylene in tomatoes can boost the biosynthesis of carotenoids. Previously, Kaewklin et al. [87] created ethylene-scavenging films containing chitosan and nano-TiO_2_ to safeguard the quality of Korean cherry tomatoes and increase their storage life by 14 days. The films demonstrated ethylene photodegradation that could delay ripening and extend the shelf life of all the most often consumed fresh produce. Another study found that using HNTs with a strong ethylene absorptivity capability slowed the crumbling and ripening of tomatoes [9]. The effectiveness of many ethylene scavengers, such as palladium-supported nano zeolite, KMnO_4_-supported nano zeolite, salicylic acid, and 1-methyl cyclopropane on maintaining the post-harvest quality of tomato fruits was recently assessed. The palladium-supported nano zeolite was found to be a technique that aids the extended storage life and preserves the quality of the tomato during storage. The palladium-supported nano zeolite exhibited an extra 5% positive effect on lycopene and phenolic content, fruit firmness, polygalacturonase activity, and weight reduction [109].

Mango fruit is subject to anthracnose and is also highly perishable. A paper coated with active ingredients is described as a possible packaging material for mango fruit that decreases anthracnose and slows the ripening process. Jaimun and Sangsuwan [110] established an ethylene scavenging paper for mango packaging with vanillin–chitosan coating with varying zeolite or activated carbon concentrations at 0 to 0.4% (*w*/*v*) within bleached paper. They wrapped commercial mango fruit (Nam Dok Mai) with active-coated paper for 30 days to examine the quality change. Weight loss, firmness, titratable acidity, and total soluble solids were much lower in mango fruits wrapped in zeolite paper (0.2%) than in uncoated papers. Similarly, the surface color of those covered in zeolite paper (0.2%) was significantly reduced.

## 8. Safety Issues and Economic Concerns

Although technological advances in ethylene scavenging packages might enhance the freshness and the security of fruit and vegetables, safety issues must be considered. Unexpected safety concerns include active substance transmission from packaging to food products, inadvertent active material leakage from a sachet to food, and human consumption of active materials. The ethylene scavenging packaging should be labelled on product packages to avoid customer misuse and misinterpretation [8]. Aside from that, with any food preservation technique, the packaging’s efficacy and potential to fulfil the ethylene scavenging function can create safety concerns. Before any unique material package design can be successfully manufactured and used by the packaging and food sectors, the applications must be examined for usability, stability, and conformity with traditional equipment and forming procedures [111,112]. The current food producer and customer demand for clean food labelling presents a challenge for releasing systems because substances released from food packaging that have a technical influence on the food must be reported in the ingredient description on package labels [13].

Health and environmental issues influence consumer perceptions of active packaging and uncertainty about emerging innovations due to inadequate education [113]. In this era, fresh produce has a lower cost than other food items, yet fresh produce has a higher final price due to the high packing cost. Commercializing new ethylene scavenging packaging may increase individual product unit costs and pricing, particularly in the initial phases of the product launches, indirectly influencing the behavior and acceptability of consumers [114]. As a result, novel packaging in the fresh produce market should be justified based on adequate expense analysis [13]. In addition, research and development of ethylene scavenging packaging materials could pave the way for lower final product costs without sacrificing food shelf life. Furthermore, designing functional materials can sustain their original mechanical and barrier properties. Simultaneously, this proves that active agents for the scavenging process during the entire logistics division, storage, and handling as packaging materials is unquestionably the most significant challenge for ethylene scavenging packaging. Technology transfer, production rescale, food safety regulations, conservational considerations, and momentous customer acceptability are all significant challenges [104,115].

## 9. Conclusions

Active packaging with ethylene scavengers is a viable alternative for the time being for ethylene-sensitive fresh produce, which has been proven to increase the product’s shelf life while preserving physical quality and freshness. The request for synthetic preservatives and plastics to be replaced is possible due to the simultaneous application of multiple ethylene scavenging systems in the film structure while protecting food products from excess ethylene. It is critical to understand the physical properties of fruit and vegetables before selecting an ethylene scavenging device or a packaging method for ethylene gas control. Conversely, the acceptance on ethylene scavenging packaging and cost-effectiveness for fresh produce growers, processors, and consumers will decide its evolution and implementation. Consequently, more attention should be devoted to settling the technical hurdles and expenses associated with these innovations, which have been the main barriers to broader adoption and the emergence of new industrial uses for ethylene scavenging materials in the food manufacturing industry sectors.

## Figures and Tables

**Figure 1 membranes-12-00117-f001:**
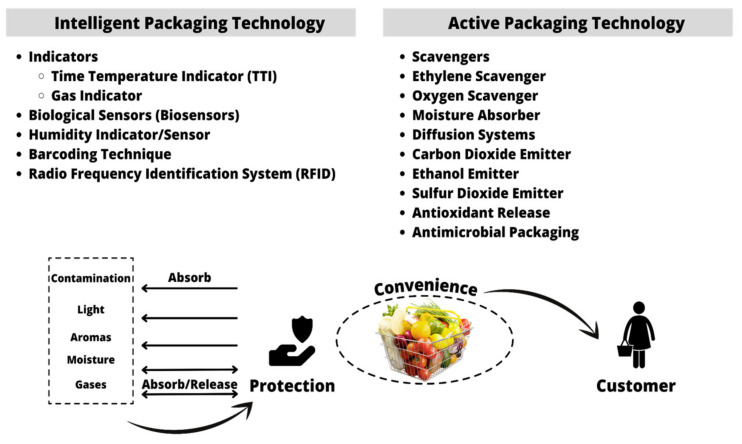
The mechanism of intelligent and active packaging in the food industry.

**Figure 2 membranes-12-00117-f002:**
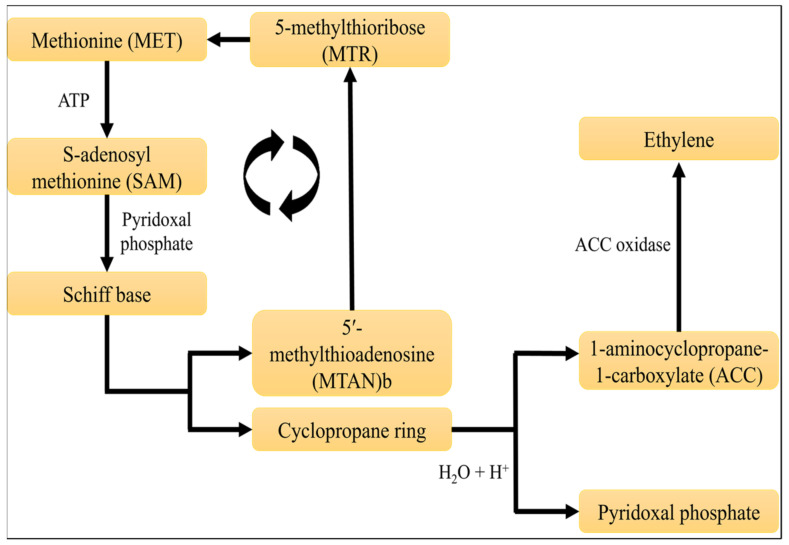
The biosynthesis mechanism of ethylene production.

## Data Availability

Not applicable.

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
