# Peer review of "The Emergence and Impact of Ethylene Scavengers Techniques in Delaying the Ripening of Fruits and Vegetables"

_membranes, 2022, doi:10.3390/membranes12020117_

Round 1

Reviewer 1 Report

The manuscript entitled "Emerging of ethylene scavengers techniques in delaying the ripening of fresh fruits and vegetables" is a very well-written manuscript and it will be interesting to those who are working in the area of active packaging. I strongly recommend the manuscript after minor revision in the English tupo correction 

  • keywords should be specific remove Keywords: "Postharvest handling"
  • In the Introduction section, recent references on ethylene scavengers should be added author can refer this recent article https://doi.org/10.1016/j.indcrop.2021.113752
  • In section, ethylene scavenging mechanism should be  separate from paragraph 

Author Response

  1. The manuscript entitled "Emerging of ethylene scavengers techniques in delaying the ripening of fresh fruits and vegetables" is a very well-written manuscript and it will be interesting to those who are working in the area of active packaging. I strongly recommend the manuscript after minor revision in the English typo correction.
  • Thank you for the suggestion. The manuscript has been checked and revise carefully to avoid the typo.

  1. keywords should be specific remove Keywords: "Postharvest handling"
  • The keywords has been specify as suggested. Also, “postharvesrt handling” has been removed.

  1. In the Introduction section, recent references on ethylene scavengers should be added author can refer this recent article https://doi.org/10.1016/j.indcrop.2021.113752
  • The references has been included as number 10 in the introduction section.

  1. In section, ethylene scavenging mechanism should be separate from paragraph
  • The section has been updated as shown in 3.2. Ethylene scavenging mechanism

Reviewer 2 Report

The manuscript was well and fluently written but similar reviews are available as like as " A Review on the Modified Atmosphere Preservation of Fruits and Vegetables with Cutting-Edge Technologies " or "Ethylene-removing packaging: Basis for development and latest advances" so I think It seems better to use and add new scientific contexts as like as " impact of using this type of packaging on sustainable economy

Author Response

The manuscript was well and fluently written but similar reviews are available as like as " A Review on the Modified Atmosphere Preservation of Fruits and Vegetables with Cutting-Edge Technologies " or "Ethylene-removing packaging: Basis for development and latest advances" so I think It seems better to use and add new scientific contexts as like as " impact of using this type of packaging on sustainable economy.

  • The context has been added in the manuscript and in the title “Emerging and impact of ethylene scavengers techniques in delaying the ripening of fruits and vegetables”

Reviewer 3 Report

This is an up-to-date review  study about the importance of ethylene in postharvest handling as well as the use of emergent ethylene scavengers.

Author Response

Reviewer #3:

General Comments:

This review is providing an up-to-date discussion and complete overview about the importance of ethylene and ethylene scavenging in postharvest technology. Its importance is eminent. Some minor modifications should be introduced that will be specified in the following section.

  • The minor modification has been done in the manuscript by following the specific comments and suggestions.

Specific Comments:

  1. Abstract (L.12-29): No information has been provided about the importance of capture of ethylene. Although these points have been stated in “Highlights”, these might be given in the Abstract as well.
  • The importance of capturing ethylene has been added in the abstract

  1. 16: Please, modify the spelling of the terms “living things” that seems to be a contradiction.
  • The terms has been revised

  1. 19: The use of the terms “ever-growing global population” should be reconsidered.
  • The terms has been revised

  1. 45-46: Deterioration of quality concept seems to include deterioration of physical appearance.
  • The sentence has been revised

  1. 47-48: Degradation of chlorophylls is a chemical microscopic phenomenon, while chlorophyl structure softening seems to be a physical macroscopic phenomenon. Softening is normally related to pectin degradation, amongst others.
  • The sentence has been revised

  1. 94-95: Check spelling.
  • The spelling has been revised

  1. 110-111: Check spelling of “absorbing and diffusion different gases”
  • The spelling has been revised

  1. 123-124: What is the meaning of “factors”?
  • The sentence has been revised

  1. Figure 1: Do you mean “selective diffusion systems”?
  • It is a diffusion systems for various materials like carbon dioxide, oxygen, and ethanol

  1. 154-156: Why is this step critical? Please, give an explanation.
  • The sentence has been revised

  1. L167: Please, check concentration value of 0.1 L/L.
  • The unit has been revised.

  1. 282-283: Please, give more details about the second aspect (bacterial growth).
  • The information has been added in the manuscript.

  1. 307: What is the meaning of “duo”?
  • The typo has been revised

  1. 315-316: The term “Stone” is repeated.
  • The term has been removed.

  1. 356-358: KMnO4 is toxic. Is its use allowed in food products? L.359-371: Is there any regulation of use of potassium permanganate in food products?
  • It is toxic and does not have FDA approved to be used in food product. It is not safe to consume.

  1. 502: The ethylene generation was lower.... Some value should be given.
  • The value has been added accordingly.

  1. 526-530: I suppose that the exposure of TiO2 to UV light and the formation of ROS depends on the concentration of antioxidant compounds in fruit and vegetables. Is there any information about this topic?
  • The information has been added in the manuscript.

  1. 558: Use “moisture” instead of “relative humidity”.
  • The sentence has been updated

  1. L566: Are there any sensors to determine sachets saturation with ethylene?
  • Currently, the author did find any sensor to determine sachets saturation with ethylene. It is a good idea for the research innovation to come out the sachet’s ethylene scavengers.

  1. 580: Check units or vales.
  • The unit has been updated

  1. 717-963: Uniform reference citation.
  • The reference has been updated

  1. 825: Incomplete reference.
  • The reference has been updated